# Free Vibration Analysis of Porous Functionally Graded Material Plates with Variable Thickness on an Elastic Foundation Using the R-Functions Method

Lidiya Kurpa [1], Francesco Pellicano [2] (ID), Tetyana Shmatko [3] (ID) and Antonio Zippo [2],*(ID)

1 Department of Applied Mathematics, National Technical University "KhPI", Kyrpychova Str. 2, 61002 Kharkiv, Ukraine

2 Department of Engineering "Enzo Ferrari", Centre InterMech MoRe, University of Modena and Reggio Emilia, Via P. Vivarelli 10, 41124 Modena, Italy; francesco.pellicano@unimore.it

3 Department of Higher Mathematics, National Technical University "KhPI", Kyrpychova Str. 2, 61002 Kharkiv, Ukraine; ktv_ua@yahoo.com

* Correspondence: antonio.zippo@unimore.it

**Abstract:** Free vibrations of porous functionally graded material (FGM) plates with complex shapes are analyzed by using the R-functions method. The thickness of the plate is variable in the direction of one of the axes. Two types of porosity distributions through the thickness are considered: uniform (even) and non-uniform (uneven). The elastic foundation is defined by two parameters (Winkler and Pasternak). To obtain the mathematical model of the problem, the first-order shear deformation theory of the plate (FSDT) is used. The effective material properties in the thickness direction are modeled by means of a power law. Variational Ritz's method joined with the R-functions theory is used for obtaining a semi-analytical solution of the problem. The approach is applied to a number of case studies and validated by means of comparative analyses carried out on rectangular plates with a traditional finite element approach. The proof of the efficiency of the approach and its capability to handle actual engineering problems is fulfilled for FGM plates having complex shapes and various boundary conditions. The effect of different parameters, such as porosity distribution, volume fraction index, elastic foundation, FGM types, and boundary conditions, on the vibrations is studied.

**Keywords:** functionally graded material; R-function method; elastic foundation

## 1. Introduction

FGMs represent modern materials that are going to be widely used in many industries, with applications in aircraft and rockets, ships, nuclear reactors, and a number of components used in mechanical engineering. The intensive use of functionally graded materials has led to the need for a thorough study of their behavior during operation, considering such characteristics as porosity, elastic foundation, and varying plate or shell thickness. In this regard, many scientists focused their investigations on both theories related to the development of mathematical models [1–11] and experiments [12,13].

It can be noted that porosity can occur in functionally graded structures during the manufacturing process. So, many researchers took into account the influence of porosity while they were investigating mechanical, thermal, and other characteristics of FGM structures. Kim et al. [14] used the classical and first-order shear deformation theory to investigate buckling, bending, and free vibration characteristics of porous FG plates. Higher-order shear deformation theory (HSDT) was applied by Cong et al. [15] for analytical modeling of the buckling and post-buckling behavior of porous FGM plates subjected to thermal and mechanical loads. Zur and Jankowski [16] carried out a multiparametric investigation of the free vibration behavior of circular porous FGM plates using the classical plate theory. Li et al. [17] considered a semi-analytical approach to investigate the free vibration behavior of porous FG cylindrical shells. Wang and Wu [18] examined the free

vibration characteristics of porous FG cylindrical shells with different boundary conditions by applying the sinusoidal shear deformation theory (SSDT). In the last few decades, functionally graded materials have been increasingly applied to nanocomposite structures in thermal and magnetic environments [19].

The literature on the vibration analysis of FGM plates resting on elastic foundations has been enriched by many scientific contributions in the last few years; the elastic foundation model based on the Winkler and Pasternak interaction has been widely applied: the Winkler model was developed for railroad tracks, and the Pasternak model introduced a new parameter to include the spring displacement in the longitudinal and lateral directions. Yang and Shen [20] conducted a vibration analysis of an initially stressed FGM plate resting on an elastic foundation; they used a simple power law for material gradation with clamped boundary conditions. Amini et al. [21] carried out a three-dimensional vibration analysis of the FGM plate resting on a Winkler foundation; Chebyshev polynomials and the Ritz method were applied to obtain the vibration modes. Results for free vibration and buckling analysis of an S-FGM (Sigmoid FGM) sandwich plate supported on an elastic foundation were reported by Singh and Harsh [22]. Malekzaden and Karami [23] studied the free vibration behavior of a homogeneous linearly varying thick plate resting on an elastic foundation using differential quadrature methods. Investigations on porous FGM structures with even and uneven distributions of porosity have been observed in the last few years. Rezaei and Saidi [24,25] analyzed the free vibration and flexural response of porous plates with different boundary conditions. Zenkour [6] investigated the static response of porous FGM single-layered and sandwich plates using a quasi-3D shear deformation theory. Trinh et al. [26] examined the effect of evenly distributed and unevenly distributed porosities on the dynamic behaviors of FG cylindrical, spherical, and hyperbolic paraboloid shells by means of the FSDT.

Nguen et al. [27] used the first-order shear deformation theory for deriving theoretical formulations and illustrating the nonlinear response of FG porous plates under thermal and mechanical loads supported by Pasternak's model of an elastic foundation. Evenly and unevenly distributed porosities were included in a distribution law for the calculation of the effective properties of FGM plates. Thrin et al. [28] developed a three-variable refined shear deformation theory to investigate the free vibration and bending behavior of porous FG doubly curved shallow shells exposed to uniform and sinusoidal pressure. Two porosity types influence parameters that influence the material properties and structure behaviors in different aspects. Kumar et al. [29] applied the first-order shear deformation theory for the presentation of a displacement model of the kinematic equations for tapered, porous FGM plates with variable thickness resting on a two-parameter elastic foundation. The solutions for constant and varying thick plates were investigated. Vinh and Huy [30] examined the static bending, free vibration, and buckling of FG sandwich plates with porosity using the finite element model based on a hyperbolic shear deformation theory. Most of the works cited above and the most of scientific literature dealt with rectangular plates. Balak et al. [31] studied the dynamic behavior of an elliptical multilayer plate with a saturated porous filler resting on an elastic foundation; these authors considered the case of the face sheet layers being piezoelectric. To solve the problem, the authors applied the first-order shear deformation theory and Galerkin's method.

Based on the above literature, it seems that many studies were conducted for the vibration analysis of FGM plates. But investigations in the field of porous FGM plates and shells with variable thickness resting on an elastic foundation are in demand now and are still limited. In particular, there is a need to investigate FGM plates and shells with complex planform and different boundary conditions.

In Ref. [32], nonlinear dynamics of spherical caps were numerically investigated, with an exploration of chaos-induced symmetry breaking. Nonlinear random vibrations of circular cylindrical shells were experimentally investigated in Ref. [33]. Modal localizations due to small imperfections in circular cylindrical shells were numerically investigated in Ref. [34]. A complex dynamic scenario of shells in contact with a non-Newtonian fluid was experimentally investigated in Ref [35].

In the present paper, the authors propose an approach combining the R-functions theory and variational Ritz method for studying the free vibration behavior of porous FGM plates of different forms with a hole of complex geometry. These plates are resting on an elastic foundation. The method allows the construction of admissible functions in an analytical form and the consideration of different boundary conditions for the hole and outside border. The thickness of the FGM plate can be varied using a linear or nonlinear law. Moreover, FEM analyses have been carried out with COMSOL Multiphysics in order to investigate the advantages and disadvantages of finite element modeling and boundary condition sensitivity and to carry out comparisons with the analytical method.

## 2. Formulation of the Problem

Consider a porous plate on an elastic foundation with variable thickness. Assume that a plate (Figure 1A) is made of functionally graded material (FGM), namely a mixture of ceramics (top of the plate) and metal (bottom). The distribution law of thickness for the general case is shown in Figure 1B. The plate may have an arbitrary shape.

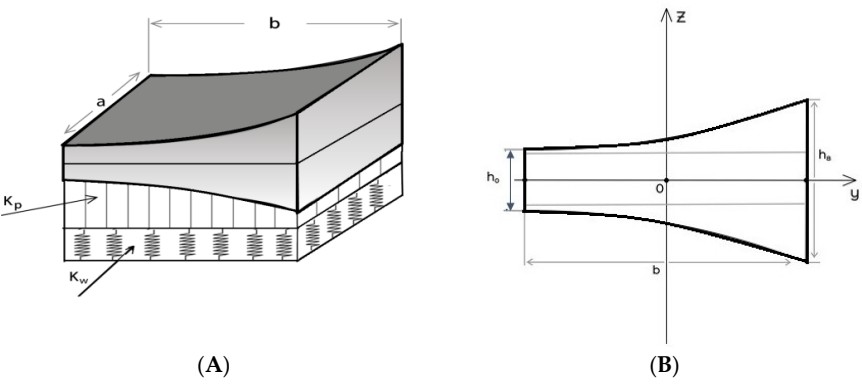

|        |        |
|:------:|:------:|
|  (A)   |  (B)   |

**Figure 1.** (**A**) FGM plate on elastic foundation with variable thickness; (**B**) distribution law of thickness for general case.

Two porosity distribution types are studied: even (Figure 2a) and uneven (Figure 2b). The effective material properties through the thickness such as Young's modulus $E$ and mass density $\rho$ can vary with a power law for an FGM with the porosity distribution factor $\alpha$ [24–26]:

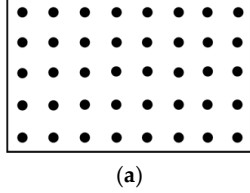
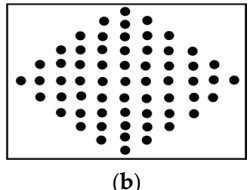

|        |        |
|:------:|:------:|
|  (a)   |  (b)   |

**Figure 2.** (**a**) Even distribution of porosity; (**b**) uneven distribution of porosity.

Type I (even) is given as follows:

$$P_{ef}(z) = (P_m(T) - P_c(T))\left(\frac{z}{h} + \frac{1}{2}\right)^p + P_c(T) - \frac{\alpha}{2}(P_c(T) + P_m(T)) \tag{1}$$

Type II (uneven) is presented as follows:

$$P_{ef}(z) = (P_m(T) - P_c(T))\left(\frac{z}{h} + \frac{1}{2}\right)^p + P_c(T) - \frac{\alpha}{2}(P_c(T) + P_m(T))\left(1 - 2\frac{|z|}{h(x,y)}\right) \tag{2}$$

Here, $p$ is the positive volume fraction index (gradient index). The mechanical characteristics of ceramics $P_c$ and metal $P_m$ depend on temperature $T$. This dependence is defined by known formulas presented in Refs. [6–8].

Equations (1) and (2) represent the general formulas for the determination of the elastic modulus $E$ and the density $\rho$ of the composite.

The stress and strain resultants in matrix form are as follows:

$$\{N\} = [A]\{\varepsilon^0\} + [B]\{\chi\}, \ \{M\} = [B]\{\varepsilon^0\} + [D]\{\chi\}, \ \{Q\} = K_s A_{33}\{\gamma_{13}, \gamma_{23}\}^T \quad (3)$$

where $\{N\} = \{N_{11}, N_{22}, N_{12}\}^T$ are forces per unit edge length in the middle surface of a plate, $\{M\} = \{M_{11}, M_{22}, M_{12}\}^T$ are bending and twisting moments per unit edge length, and $Q = (Q_x, Q_y)^T$ are transverse resultants; the coefficient $K_s^2$ denotes the shear correction factor. The components of the vectors $\{\varepsilon^0\} = \{\varepsilon^0_{11}, \varepsilon^0_{22}, \gamma^0_{12}\}^T$ and $\{\chi\} = \{\chi_{11}, \chi_{22}, \chi_{12}\}^T$ are defined in the following way:

$$\varepsilon^0_{11} = u_{,x} \ \varepsilon^0_{22} = v_{,y} \ \gamma^0_{12} = u_{,y} + v_{,x}, \gamma_{13} = w_{,x} + \psi_x, \ \gamma_{23} = w_{,y} + \psi_y, \quad (4)$$

$$\chi_{11} = \psi_{x,x}, \ \chi_{22} = \psi_{y,y}, \ \chi_{12} = \psi_{x,y} + \psi_{y,x}$$

where $u$ and $v$ are middle surface displacements along the axes $Ox$ and $Oy$, respectively; $w$ is the transverse deflection of the plate along the axis $Oz$; and $\psi_x$, $\psi_y$ are angles of rotations of the normal to the middle surface about the axes $Oy$ and $Ox$. Matrices $[A]$, $[B]$, $[D]$ have the following form:

$$[A] = \begin{bmatrix} A_{11} & A_{12} & 0 \\ A_{12} & A_{22} & 0 \\ 0 & 0 & A_{66} \end{bmatrix}, [B] = \begin{bmatrix} B_{11} & B_{12} & 0 \\ B_{12} & B_{22} & 0 \\ 0 & 0 & B_{66} \end{bmatrix}, [D] = \begin{bmatrix} D_{11} & D_{12} & 0 \\ D_{12} & D_{22} & 0 \\ 0 & 0 & D_{66} \end{bmatrix}, \quad (5)$$

Elements of the matrixes $[A]$, $[B]$, $[D]$ are calculated as follows:

$$([A], [B], [D]) \int_{-\frac{h(x,y)}{2}}^{\frac{h(x,y)}{2}} E_{eff}(z)[C]\left(1, z, z^2\right)dz, \ [C] = \frac{1}{1-v^2}\begin{bmatrix} 1 & v & 0 \\ v & 1 & 0 \\ 0 & 0 & \frac{1-v}{2} \end{bmatrix} \quad (6)$$

Assuming that Poisson's ratio $v$ is constant and the same for metal and ceramics, we can calculate exactly the values of the elements $A_{ij}$, $B_{ij}$, $D_{ij}(i, j = 1, 2, 6)$ of matrices (5):

$$A_{11} = A_{22} = \frac{1}{1-v^2}\left(\frac{E_m - E_c}{p+1} + E_c - \frac{\alpha}{2}(E_c + E_m) + \delta\left(\frac{\alpha(E_c + E_m)}{4}\right)\right)h(x,y), \quad (7)$$

$$A_{12} = vA_{11}, \ A_{66} = A_{44} = A_{55} = \frac{A_{11}(1-v)}{2}$$

$$B_{11} = B_{22} = \frac{(E_m - E_c)p}{(1-v^2)(p+1)(p+2)}\frac{h^2(x,y)}{2}, \ B_{12} = vB_{11}, \ B_{66} = B_{44} = B_{55} = \frac{B_{11}(1-v)}{2} \quad (8)$$

$$D_{11} = D_{22} = \frac{h^3(x,y)}{1-v^2}\left(\frac{E_c}{12} - \frac{\alpha}{24}(E_c + E_m) + (E_m - E_c)\left(\frac{1}{p+3} - \frac{1}{p+2} + \frac{1}{4(p+1)}\right) + \delta\left(\frac{\alpha(E_c + E_m)}{32}\right)\right),$$

$$D_{12} = vD_{11}, \ D_{66} = D_{44} = D_{55} = \frac{D_{11}(1-v)}{2} \quad (9)$$

where $\alpha$ is the porosity distribution factor; the indicator $\delta$ is the tracing, and this constant is equal to 0 for the even porosity and 1 for the uneven porosity.

The influence of the elastic foundation is taken into account as the reaction–deflection relation of Pasternak using the following formula [4,27]:

$$p_0 = K_w w - K_P \nabla^2 w \quad (10)$$

where $\nabla^2 w = \frac{\partial^2 w}{\partial x^2} + \frac{\partial^2 w}{\partial y^2}$, and $K_w$, $K_P$ are the Winkler foundation stiffness and the shear stiffness of the Pasternak foundation, respectively.

The equations of motion for the FGM plates are the following:

$$\frac{\partial N_{11}}{\partial x} + \frac{\partial N_{12}}{\partial y} = I_0 \frac{\partial^2 u}{\partial t^2} + I_1 \frac{\partial^2 \psi_x}{\partial t^2};$$

$$\frac{\partial N_{22}}{\partial y} + \frac{\partial N_{12}}{\partial x} = I_0 \frac{\partial^2 v}{\partial t^2} + I_1 \frac{\partial^2 \psi_y}{\partial t^2};$$

$$\frac{\partial Q_x}{\partial x} + \frac{\partial Q_y}{\partial y} - \left(K_w w - K_P \nabla^2 w\right) = I_0 \frac{\partial^2 w}{\partial t^2};$$

$$\frac{\partial M_{11}}{\partial x} + \frac{\partial M_{12}}{\partial y} - Q_x = I_2 \frac{\partial^2 \psi_x}{\partial t^2} + I_1 \frac{\partial^2 u}{\partial t^2};$$

$$\frac{\partial M_{22}}{\partial y} + \frac{\partial M_{12}}{\partial x} - Q_y = I_2 \frac{\partial^2 \psi_y}{\partial t^2} + I_1 \frac{\partial^2 v}{\partial t^2}, \tag{11}$$

where

$$(I_0, I_1, I_2) = \int_{-\frac{h(x,y)}{2}}^{\frac{h(x,y)}{2}} \rho(z)\left(1, z, z^2\right) dz \tag{12}$$

By integrating relation (12), we obtain analytical expressions of these coefficients in the following form:

$$I_0 = \left(\frac{\rho_m - \rho_c}{p+1} + \rho_c - \frac{\alpha}{2}(\rho_c + \rho_m) + \delta\left(\frac{\alpha(\rho_c + \rho_m)}{4}\right)\right) h(x,y) \tag{13}$$

$$I_1 = \frac{(\rho_m - \rho_c)p}{(1-v^2)(p+1)(p+2)} \frac{h^2(x,y)}{2}, \tag{14}$$

$$I_2 = h^3(x,y)\left(\frac{\rho_c}{12} - \frac{\alpha}{24}(\rho_c + \rho_m) + (\rho_m - \rho_c)\left(\frac{1}{p+3} - \frac{1}{p+2} + \frac{1}{4(p+1)}\right) + \delta\left(\frac{\alpha(\rho_c + \rho_m)}{32}\right)\right) \tag{15}$$

The total potential energy of the system is expressed as

$$\Pi = U + V_e - T \tag{16}$$

where strain energy $U$, potential $V_e$ and kinetic energy $T$ in this case are defined by the following expressions:

$$U = \int_\Omega \left(N_{11}\varepsilon_{11}^{(0)} + N_{22}\varepsilon_{22}^{(0)} + N_{12}\gamma_{12}^{(0)} + M_{11}\chi_{11} + M_{22}\chi_{22} + M_{12}\chi_{12} + Q_1\gamma_{13} + Q_2\gamma_{23}\right) d\Omega \tag{17}$$

$$V_e = \frac{1}{2}\int_\Omega \left(K_w w^2 + K_p\left(\vec{\nabla w}\right)^2\right) d\Omega \tag{18}$$

$$T = \frac{1}{2}\int_\Omega \left(\left(I_0\left(\left(\frac{\partial u_0}{\partial t}\right)^2 + \left(\frac{\partial v_0}{\partial t}\right)^2 + \left(\frac{\partial w_0}{\partial t}\right)^2\right) + 2I_1\left(\frac{\partial \psi_x}{\partial t}\frac{\partial u_0}{\partial t} + \frac{\partial \psi_y}{\partial t}\frac{\partial v_0}{\partial t}\right) + I_2\left(\left(\frac{\partial \psi_x}{\partial t}\right)^2 + \left(\frac{\partial \psi_y}{\partial t}\right)^2\right)\right)\right) d\Omega \tag{19}$$

## 3. Solution Method

To solve this problem, we use the Ritz method, which is an effective method with some drawbacks in application. For example, the main difficulty, arising in the case of complex geometry, is the construction of admissible functions; such a problem can be solved by using the R-functions theory. The application of the Ritz method in combination with the R-functions theory (RFM) allows one to represent the unknown solution in an analytical form. This is a great advantage of this method compared to other numerical methods. The R-functions theory offers approaches to the construction of so-called solution structures [36–39]. Those solution structures are the base for constructing the system of admissible functions. For example, let us construct the solution structures and set of admissible functions for the clamped plate shown in Figure 3a.

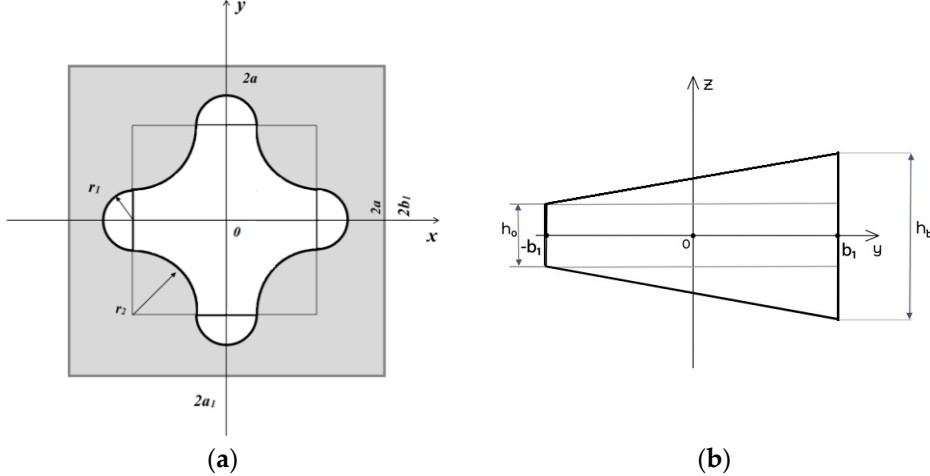

**Figure 3.** (**a**) Planform of plate with complex hole; (**b**) distribution law of thickness for plate with complex geometry.

The boundary conditions for a plate clamped on the whole border including the cut (CL-CL) are

$$w = 0, \ u = 0, \ v = 0, \ \psi_x = 0, \ \psi_y = 0, \ \forall (x,y) \in \partial\Omega \tag{20}$$

For these boundary conditions, the solution structure has the following form:

$$w = \omega\Phi_1, \ u = \omega\Phi_2, \ v = \omega\Phi_3, \ \psi_x = \omega\Phi_4, \ \psi_y = \omega\Phi_5. \tag{21}$$

The functions $\omega(x,y)$ are constructed using the R-functions theory and satisfy the following conditions:

$$\omega(x,y) = 0, \ \forall (x,y) \in \partial\Omega,$$
$$\omega(x,y) > 0, \ \forall (x,y) \in \Omega.$$

In the given case, the functions $\omega(x,y)$ have the following form:

$$\omega(x,y) = (f_1 \wedge_0 f_2) \wedge_0 \omega_{cut} \tag{22}$$

$$\omega_{cut}(x,y) = -(((f_3 \vee_0 f_4) \vee_0 f_{12}) \wedge_0 ((f_5 \vee_0 f_6) \vee_0 f_{11}) \wedge_0 ((f_7 \wedge_0 f_8) \wedge_0 (f_9 \wedge_0 f_{10}))) \tag{23}$$

where symbols $\wedge_0, \vee_0$ denote the R-operations of the $R_0$-system [36], which have the following form:

- $x_1 \wedge_0 x_2 \equiv x_1 + x_2 - \sqrt{x_1^2 + x_2^2}$ is the R-conjunction, which describes the intersection of the domains;
- $x_1 \vee_0 x_2 \equiv x_1 + x_2 + \sqrt{x_1^2 + x_2^2}$ is the R-disjunction, which describes the union of the domains.

Functions $f_i, \ i = \overline{1, 12}$ in relations (22) and (23) are defined as follows:

$$f_1 = (a_1^2 - x^2)/2a_1 \geq 0, \ f_{11} = (a^2 - x^2)/2a \geq 0,$$
$$f_2 = (b_1^2 - y^2)/2b_1 \geq 0, \ f_{12} = (a^2 - y^2)/2a \geq 0,$$
$$f_3 = \left(r_1^2 - x^2 - (y - a)^2\right)/2r_1 \geq 0, \ f_4 = \left(r_1^2 - x^2 - (y + a)^2\right)/2r_1 \geq 0,$$
$$f_5 = \left(r_1^2 - (x - a)^2 - y^2\right)/2r_1 \geq 0, \ f_6 = \left(r_1^2 - (x + a)^2 - y^2\right)/2r_1 \geq 0, \tag{24}$$
$$f_7 = \left((x - a)^2 + (y + a)^2 - r_2^2\right)/2r_2 \geq 0, \ f_8 = \left((x + a)^2 + (y - a)^2 - r_2^2\right)/2r_2 \geq 0,$$
$$f_9 = \left((x - a)^2 + (y - a)^2 - r_2^2\right)/2r_2 \geq 0, \ f_{10} = \left((x + a)^2 + (y + a)^2 - r_2^2\right)/2r_2 \geq 0,$$

In Formulas (21), the functions $\Phi_i(x,y)$, $\left(i = \overline{1,5}\right)$ are indefinite components of the structure solutions. These components are expanded in series on some complete system of functions $\left\{\varphi_i^{(k)}\right\}$, $(k = 1,2,3,4,5)$:

$$\Phi_1 = \sum_{i=1}^{N_1} a_i\varphi_i^{(1)}, \Phi_2 = \sum_{i=N_1+1}^{N_2} a_i\varphi_i^{(2)}, \Phi_3 = \sum_{i=N_2+1}^{N_3} a_i\varphi_i^{(3)}, \Phi_4 = \sum_{i=N_3+1}^{N_4} a_i\varphi_i^{(4)}, \Phi_5 = \sum_{i=N_4+1}^{N_5} a_i\varphi_i^{(5)}, \quad (25)$$

where $a_i$, $i = 1,2,\cdots N_5$ are indefinite coefficients.

After the substitution of expressions (25) into (21), the following representation of the sought solution can be obtained:

$$w(x,y) = \sum_{i=1}^{N_1} a_i w_i(x,y), \; u(x,y) = \sum_{i=N_1+1}^{N_2} a_i u_i(x,y), \; v(x,y) = \sum_{i=N_2+1}^{N_3} a_i v_i(x,y),$$

$$\psi_x(x,y) = \sum_{i=N_3+1}^{N_2} a_i\psi_{xi}(x,y), \; \psi_y(x,y) = \sum_{i=N_4+1}^{N_3} a_i\psi_{yi}(x,y) \quad (26)$$

The functions

$$w_i = \omega(x,y)\varphi_i^{(1)}(x,y), \; u_i = \omega(x,y)\varphi_i^{(2)}(x,y), \; v_i = \omega(x,y)\varphi_i^{(3)}(x,y),$$

$$\psi_{xi} = \omega(x,y)\varphi_i^{(4)}(x,y), \; \psi_{yi} = \omega(x,y)\varphi_i^{(5)}(x,y) \quad (27)$$

are basic functions that satisfy boundary conditions (20) for any choice of the indefinite coefficients. These coefficients are sought by the Ritz method from the condition for the corresponding functional to have a stationary point. It is easy to show that in the given case, for harmonic vibrations, this functional has the following form:

$$J = U + V_e - \lambda^2 P \quad (28)$$

where $U$, $V_e$ and $P$ are the maximum potential and kinetic energies, respectively. Let us note that expressions for $U$ and $V_e$ coincide with (17) and (18).

The maximum kinetic energy $T$ takes the following form:

$$P = \frac{1}{2}\iint_\Omega \left(I_0\left(u^2 + v^2 + w^2\right) + 2I_1\left(u\psi_x + v\psi_y\right) + I_2\left(\psi_x^2 + \psi_y^2\right)\right)dxdy \quad (29)$$

Here, $\lambda$ is a natural frequency of the harmonic vibrations in Equation (28).

## 4. Numerical Results and Discussion

### 4.1. Validation of the Approach

To prove the effectiveness of the proposed method and the accuracy of numerical results, a validation analysis was carried out for square FGM plates; different cases were considered by changing the parameters of porosity, volume exponent, elastic foundation, FGM type, and boundary conditions. The following five tests describe several comparative examples.

Test 1

Free vibrations of a simply supported FGM $Si_3N_4/SUS304$ square plate with porosity (Type 1) are considered. The mechanical characteristics of the mixture are as follows:

| | | | |
|---|---|---|---|
| $Si_3N_4$ : | $E = 322.27$ GPa, | $\nu = 0.3$, | $\rho = 2370$ kg/m$^3$; |
| $SUS304$ : | $E = 207.78$ GPa, | $\nu = 0.3$, | $\rho = 8166$ kg/m$^3$. |

This plate is porous with porosity distribution factor $\alpha = 0, 0.1, 0.2$ and changing gradient index $p = 0, 1, 2, 5, 10, 100$. Values of non-dimensional fundamental frequency $\Lambda = \lambda (2a)^2 h_0 \sqrt{\rho_c/E_c}$ for this study are presented in Table 1. A comparative analysis was carried out via comparisons with Ref. [28]. It is observed that there is an excellent agreement.

**Table 1.** Comparison of non-dimensional fundamental frequency $\Lambda = \lambda (2a)^2 h_0 \sqrt{\rho_c/E_c}$ for simply supported porous $Si_3N_4/SUS304$ FG plates ($\frac{a}{b} = 1$, $\frac{h}{2a} = 0.1$) with Ref. [28].

| $p$ | $\alpha = 0$ | | $\alpha = 0.1$ | | $\alpha = 0.2$ | |
|---|---|---|---|---|---|---|
| | **RFM** | **[28]** | **RFM** | **[28]** | **RFM** | **[28]** |
| 0 | 0.0249 | 0.0250 | 0.0241 | 0.0241 | 0.0231 | 0.0231 |
| 1 | 0.0348 | 0.0348 | 0.0347 | 0.0348 | 0.0347 | 0.0347 |
| 2 | 0.0394 | 0.0394 | 0.0399 | 0.0399 | 0.0406 | 0.0406 |
| 5 | 0.0460 | 0.0460 | 0.0477 | 0.0477 | 0.0502 | 0.0501 |
| 10 | 0.0503 | 0.0503 | 0.0531 | 0.0530 | 0.0572 | 0.0571 |
| 100 | 0.0567 | 0.0567 | 0.0613 | 0.0614 | 0.0688 | 0.0688 |

Test 2

An investigation of the free vibration of a square FGM plate on an elastic foundation made of aluminum and alumina (Al/Al$_2$O$_3$) without porosity ($\alpha = 0$) is conducted. The mechanical characteristics are as follows:

| Al: | $E = 70$ GPa, | $\nu = 0.3$, | $\rho = 2707$ kg/m$^3$ |
|---|---|---|---|
| Al$_2$O$_3$: | $E = 380$ GPa, | $\nu = 0.3$, | $\rho = 3800$ kg/m$^3$ |

Non-dimensional fundamental frequencies $\Lambda = \lambda (2a)^2 h_0 \sqrt{\rho_m/E_m}$ were obtained for $K_s = \frac{\pi^2}{12}$ using the FSDT. The elastic stiffnesses of Winkler and Pasternak foundations are defined as follows:

$$\overline{K}_W = \frac{K_W h_0^3}{(2b)^4 12(1 - \nu_m \nu_c)}, \overline{K}_P = \frac{K_P h_0^3}{(2b)^2 12(1 - \nu_m \nu_c)}$$

Four different values of the volume exponent index $p$ = 0, 1, 2, 5 are taken. Table 2 shows a comparison with results from Refs. [29,40]; an additional comparison with the finite element method (FEM) using COMSOL Multiphysics V6.2 is presented in Table 2.

**Table 2.** Comparison of non-dimensional fundamental frequency $\Lambda = \lambda (2a)^2 h_0 \sqrt{\rho_m/E_m}$ for simply supported (without porosity) FG (Al/Al$_2$O$_3$) plates ($\frac{a}{b} = 1$, $\frac{h_0}{2a} = 0.05$, $K_s = \frac{\pi^2}{12}$) on elastic foundation with Refs. [29,40] and FEM.

| $K_w$ | $K_p$ | Method | $p = 0$ | $p = 1$ | $p = 2$ | $p = 5$ |
|---|---|---|---|---|---|---|
| 0 | 0 | RFM | 0.0291 | 0.0222 | 0.0202 | 0.0191 |
| | | [29] | 0.0291 | 0.0222 | 0.0202 | 0.0191 |
| | | [40] | 0.0291 | 0.0222 | 0.0202 | 0.0191 |
| | | FEM | 0.0286 | 0.0219 | 0.0199 | 0.0187 |
| 0 | 100 | RFM | 0.0406 | 0.0378 | 0.0374 | 0.0377 |
| | | [29] | 0.0406 | 0.0378 | 0.0374 | 0.0377 |
| | | [40] | 0.0406 | 0.0378 | 0.0374 | 0.0377 |
| | | FEM | 0.0384 | 0.0352 | 0.0347 | 0.0347 |
| 100 | 0 | RFM | 0.0298 | 0.0233 | 0.0214 | 0.0205 |
| | | [29] | 0.0298 | 0.0233 | 0.0214 | 0.0205 |
| | | [40] | 0.0298 | 0.0233 | 0.0214 | 0.0205 |
| 100 | 100 | RFM | 0.0411 | 0.0384 | 0.0381 | 0.0384 |
| | | [29] | 0.0411 | 0.0384 | 0.0381 | 0.0384 |
| | | [40] | 0.0411 | 0.0384 | 0.0381 | 0.0384 |

A comparison of the results obtained shows that the deviation of the frequencies calculated using the FEM is over 7% for the case of Pasternak's elastic foundation, while the results obtained using the RFM are in good agreement with the values from Refs. [29,40].

Test 3

Verification of the proposed method for square isotropic plates with various thicknesses $a/h_0$ = 5, 10, 100 and different boundary conditions is performed. The following conditions are considered: The plate is simply supported on the whole boundary (SSSS); the plate is completely clamped on the whole boundary (CCCC); the plate has two opposite sides clamped and two other sides simply supported (CSCS); the plate has two adjacent sides clamped and two other adjacent sides simply supported (CCSS). The thickness of the plate varies linearly in the $y$-direction [29]. Taper ratio $\beta$ is set equal to 0.25. A comparison of the present fundamental frequencies $\Lambda = \lambda (2a)^2 \sqrt{\rho_m/E_m}/h_0$ with Refs. [27,31] is presented in Table 3.

**Table 3.** Comparison of the fundamental frequencies $\Lambda = \lambda (2a)^2 \sqrt{\rho_m/E_m}/h_0$ for isotropic square tapered plates, $\beta$ = 0.25, with different boundary conditions and thicknesses with Refs. [27,31].

| $a/h_0$ | Method | SSSS | CCCC | CSCS | CCSS |
|---------|--------|------|------|------|------|
| 100 | RFM | 22.176 | 40.310 | 35.578 | 32.473 |
| | [31] | 22.164 | 40.309 | - | 32.441 |
| | [27] | 22.308 | 41.998 | - | 32.306 |
| 10 | RFM | 21.209 | 35.668 | 31.909 | 29.273 |
| | [31] | 21.224 | 35.65 | - | 29.339 |
| | [27] | 21.351 | 38.098 | - | 29.856 |
| 5 | RFM | 19.012 | 28.192 | 25.645 | 23.590 |
| | [31] | 19.057 | 28.154 | | 23.887 |
| | [27] | 19.1493 | 30.976 | | 25.043 |

Table 4 shows the fundamental frequencies $\Lambda = \lambda (2a)^2 \sqrt{\rho_m/E_m}/h_0$ given by the present approach for isotropic square simply supported tapered plates with different thicknesses $a/h$ = 5, 10, 100 and values of ratio tapers, $\beta$ = 0.5 and $\beta$ = 1, compared with the results of Refs. [29,41] and with finite element method (COMSOL Multiphysics) for the case $a/h_0 = 100$.

**Table 4.** Comparison of the fundamental frequencies $\Lambda = \lambda (2a)^2 \sqrt{\rho_m/E_m}/h_0$ for isotropic square simply supported square tapered plates with different thicknesses and taper ratios with Refs. [29,41].

| Taper Ratio $\beta$ | Method | $a/h_0$ = 100 | $a/h_0$ = 10 | $a/h_0$ = 5 |
|---------------------|--------|---------------|--------------|-------------|
| 0.5 | RFM | 24.554 | 23.258 | 20.450 |
| | [41] | 24.543 | 23.282 | 20.518 |
| | [29] | 25.059 | 23.728 | 20.834 |
| | FEM | 21.570 | - | - |
| 1 | RFM | 29.193 | 27.062 | 22.882 |
| | [41] | 29.184 | 27.120 | 23.031 |
| | [29] | 30.897 | 28.511 | 23.928 |
| | FEM | 25.430 | - | - |

For plates having a variable thickness, the deviation of the results obtained using the RFM does not exceed 2% compared to the results presented in Ref. [29], and the results obtained have a good agreement with the results in Ref. [41]. The results obtained using the FEM differ significantly from the results presented in Refs. [29,41], and the error is about 13%. The authors can suppose that the FEM is not very useful for this specific problem of plates with variable thickness due to the ratio between length and thickness.

Test 4

This example presents vibration analysis of square FGM plates (Al/Al$_2$O$_3$) with different boundary conditions (SSSS, CCCC, CSCS), changing gradient index $p = 3, 5$, and porosity distribution factor $\alpha = 0, 0.1, 0.2$. Two porosity distribution types (even and uneven) are considered; the taper ratio is $\beta = 0.4$. The shear deformation factor is taken as $K_s = \frac{\pi^2}{12}$. The results are shown in Table 5. Good agreement was found with Ref. [29] and with the finite element method (COMSOL Multiphysics) for the even distribution of porosity.

**Table 5.** Comparison of non-dimensional frequency parameter $\Lambda = \lambda\,(2a)^2\sqrt{\rho_m/E_m}/h_0$ for square FGM (Al/Al$_2$O$_3$) plates with different boundary conditions, gradient index values, and porosity parameters (Type I and Type II), $K_s = \frac{\pi^2}{12}$, $a/h_0 = 20$, $\beta = 0.4$, with Ref. [29] and FEM.

| | $p$ | Method | $\alpha = 0$ | $\alpha = 0.1$ | | $\alpha = 0.2$ | |
|---|---|---|---|---|---|---|---|
| | | | | Even | Uneven | Even | Uneven |
| SSSS | 3 | RFM | 9.4060 | 8.7741 | 9.3335 | 7.7039 | 9.2250 |
| | | [29] | 9.4611 | 8.8289 | 9.3890 | 7.7603 | 9.2819 |
| | | FEM | 9.2800 | 8.6400 | - | 7.5500 | - |
| | 5 | RFM | 9.1969 | 8.5106 | 9.1143 | 7.2713 | 8.9880 |
| | | [29] | 9.2501 | 8.5637 | 9.1687 | 7.3255 | 9.0437 |
| | | FEM | 9.2300 | 8.5300 | - | 7.2500 | - |
| CCCC | 3 | RFM | 16.7182 | 15.6250 | 16.5544 | 13.7710 | 16.3894 |
| | | [29] | 17.5125 | 16.3599 | 17.3756 | 14.4083 | 17.1745 |
| | | FEM | 16.9200 | 15.7600 | - | 13.8100 | - |
| | 5 | RFM | 16.2971 | 15.1072 | 16.4071 | 12.9621 | 15.9080 |
| | | [29] | 17.0913 | 15.8377 | 16.9332 | 13.5777 | 16.6951 |
| | | FEM | 16.8000 | 15.5000 | - | 13.2000 | - |
| CSCS | 3 | RFM | 13.5327 | 12.6407 | 13.4251 | 11.1286 | 13.2670 |
| | | [29] | 13.5449 | 12.6487 | 13.4400 | 11.1319 | 13.1480 |
| | | FEM | 13.6100 | 12.6700 | - | 11.1000 | - |
| | 5 | RFM | 13.2024 | 12.2324 | 13.0778 | 10.4829 | 12.8912 |
| | | [29] | 13.2271 | 12.2532 | 13.1069 | 10.4967 | 12.9247 |
| | | FEM | 13.5300 | 12.4800 | - | 12.1000 | - |

The comparison with the literature leads to the conclusion that the present approach is quite accurate, and the model can be considered validated for porous FGM plates of variable thickness resting on an elastic foundation. Note that in this case, the results obtained using the RFM and FEM methods are close and differ by no more than 2%. In the following section, further analyses will be presented, focused on plates having a complex geometry. In certain instances, such as when $p = 5$, alpha = 0.2, and CSCS, a discrepancy arises, which may be attributed to an edge (boundary) layer issue that eludes detection in the FEM, as indicated in Ref. [11]. In the following section further analyses will be presented, focused on plates having a complex geometry.

### 4.2. Free Vibration Analysis for Porous FGM Plates with a Complex Geometry

In actual engineering problems, professionals often face complex geometries, including, for example, holes having regular (circles or squares) or other shapes. These geometries are typically very difficult to handle with analytical or semi-analytical methods. Here, we prove that the use of the R-functions method allows the aforementioned difficulties to be overcome in an efficient way.

In this section, a porous plate with variable thickness resting on an elastic foundation with different elastic stiffnesses of Winkler and Pasternak types is investigated. The Planform of the plate with complex geometry and the distribution law of its thickness are shown in Figure 3a,b.

The geometric parameters are

$$a_1/a = 3;\ a_1/b_1 = 1;\ r_1/2a\ =\ 0.15;\ r_2/2a\ =\ 0.35;\ h_0/2a\ =\ 0.05.$$

The thickness of the plate varies linearly in the $y$-direction:

$$h(y)\ =\ h_0\left(1 + \beta\left(\frac{y+b}{2b}\right)\right),\ \beta\ =\ \frac{h_b - h_0}{h_o}. \tag{30}$$

The shear correction factor $K_s\ =\ \frac{5}{6}$ was chosen for the current analysis. The following boundary conditions were considered for the numerical experiment:

1. CL-CL—the outside boundary and hole are clamped.
2. CL-F—outside boundary is clamped, hole is free.
3. SS-CL—outside boundary is simply supported, hole is clamped.
4. F-CL—outside boundary is free, hole is clamped.

The non-dimensional natural frequency is defined as $\Lambda\ =\ \Omega\,(2a)^2\sqrt{\rho_m/E_m}/h_0$ for all cases.

Case 1

An investigation of the influence of the gradient index ($p$ = 0, 0.5, 1, 2, 5, 10) on the natural frequency for an ideal FGM plate with constant thickness and complex geometry (Figure 3a) made of Al/Al$_2$O$_3$ for four types of boundary conditions is fulfilled; the results are shown in Table 6.

**Table 6.** Effect of gradient index on natural frequency of FGM (Al/Al$_2$O$_3$) plate (Figure 3a) with different boundary conditions.

| $p$ | CL-CL | CL-F | SS-CL | F-CL |
|-----|-------|------|-------|------|
| 0 | 12.3781 | 2.7204 | 8.2491 | 1.9158 |
| 0.5 | 10.6016 | 2.3044 | 7.0379 | 1.6449 |
| 1 | 9.6044 | 2.0768 | 6.3645 | 1.4923 |
| 2 | 8.7202 | 1.8881 | 5.7809 | 1.3546 |
| 5 | 8.1172 | 1.7889 | 5.4146 | 1.2559 |
| 10 | 7.7652 | 1.7316 | 5.2021 | 1.1991 |

From this analysis, it follows that for all boundary conditions, the natural frequencies decrease with an increase in the gradient index $p$. The results obtained for plates with the boundary condition CL-CL are essentially greater than frequencies for plates with other types of boundary conditions. The smallest values of frequency and minor changes are observed for boundary condition F-CL.

Case 2

An investigation of the free vibrations of a clamped (CL-CL) FGM (Al/Al$_2$O$_3$) plate resting on an elastic foundation with various combinations of Winkler and Pasternak types and different values of the gradient index $p$ = 0, 0.5, 1, 2, 5, 10 is conducted. The non-dimensional fundamental frequencies are graphically illustrated in Figure 4.

It can be seen that the influence of the Pasternak foundation parameter exceeds that of the Winkler foundation parameter in all cases. A Pasternak foundation contains the effect of transverse shear deformation of elastic springs.

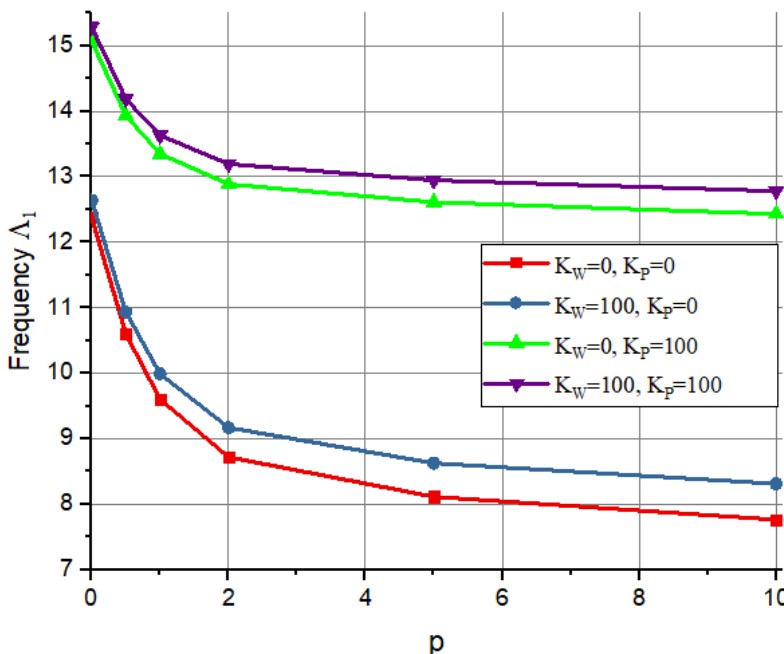

**Figure 4.** Effect of foundation stiffness $K_w$, $K_p$ on natural frequency of FGM ($Al/Al_2O_3$) clamped plate on elastic foundation.

Case 3

An investigation of the vibrations of clamped (CL-CL) FGM plates having a complex geometry (Figure 3), resting on an elastic foundation (stiffness coefficients $K_w = 100$, $K_p = 100$), for different values of the porosity parameter $\alpha = 0, 0.1, 0.2, 0.3$ is performed. The gradient index is varied as follows: $p = 0, 0.5, 1, 2, 5, 10$; three types of porosities are considered: ideal, even, and uneven. The non-dimensional fundamental frequencies are reported in Table 7. For this case, all of the natural frequencies do not increase significantly with an increase in the porosity parameter for both types—even and uneven distribution.

**Table 7.** Effect of the porosity parameter Type I and Type II on natural frequency of FGM ($Al/Al_2O_3$) clamped plate on elastic foundation ($\beta = 0, K_w = 100, K_p = 100$).

| $p$ | $\alpha = 0$ | $\alpha = 0.1$ | | $\alpha = 0.2$ | | $\alpha = 0.3$ | |
|---|---|---|---|---|---|---|---|
| | **Ideal** | **Even** | **Uneven** | **Even** | **Uneven** | **Even** | **Uneven** |
| 0 | 15.2845 | 15.6671 | 15.5264 | 16.1161 | 15.7863 | 16.6514 | 16.0661 |
| 0.5 | 14.1874 | 14.5062 | 14.4108 | 14.8842 | 14.6519 | 15.3404 | 14.9133 |
| 1 | 13.6306 | 13.8831 | 13.8305 | 14.1729 | 14.0455 | 14.5053 | 14.2773 |
| 2 | 13.1943 | 13.3658 | 13.3646 | 13.5266 | 13.5434 | 13.6234 | 13.7305 |
| 5 | 12.9451 | 13.0728 | 13.0998 | 13.1276 | 13.2553 | 12.8635 | 13.4066 |
| 10 | 12.7809 | 12.9191 | 12.9407 | 12.9787 | 13.1009 | 12.5886 | 13.2544 |

Case 4

An investigation of free vibration behavior for a clamped (CL-CL) FGM plate ($Al/Al_2O_3$) with variable thickness on an elastic foundation is performed. For this experiment, the taper ratio is varied as follows: $\beta = [0, 0.1, 0.2, 0.3]$; the porosity parameter is $\alpha = 0.2$, and the foundation stiffness coefficients are $K_w = 50$, $K_p = 100$. Two porosity distribution types are considered: even and uneven. The effect of the taper ratio $\beta$ on the natural frequency of an FGM plate is tabulated in Table 8 and shown in Figure 5.

**Table 8.** Effect of taper ratio $\beta$ and gradient index $p$ on natural frequency of clamped FGM plate on elastic foundation ($\alpha = 0.2$, Type I, II, $K_w = 50$, $K_p = 100$).

| $p$ | $\beta = 0$ | | $\beta = 0.1$ | | $\beta = 0.2$ | | $\beta = 0.3$ | |
|---|---|---|---|---|---|---|---|---|
| | Even | Uneven | Even | Uneven | Even | Uneven | Even | Uneven |
| 0 | 15.9938 | 15.6732 | 16.4946 | 16.1629 | 16.8211 | 16.4823 | 17.0981 | 16.2578 |
| 0.5 | 14.7343 | 14.5156 | 15.1997 | 14.9725 | 15.5032 | 15.2705 | 15.7629 | 15.5251 |
| 1 | 14.0044 | 13.8944 | 14.4487 | 14.3306 | 14.7388 | 14.6197 | 14.9889 | 14.8662 |
| 2 | 13.3365 | 13.3762 | 13.7609 | 13.7988 | 14.0396 | 14.0767 | 14.2821 | 14.3177 |
| 5 | 12.9157 | 13.0722 | 13.3287 | 13.4858 | 13.6052 | 13.7621 | 13.8484 | 14.0049 |
| 10 | 12.7579 | 12.9093 | 13.1689 | 13.3206 | 13.4500 | 13.5988 | 13.6981 | 13.8439 |

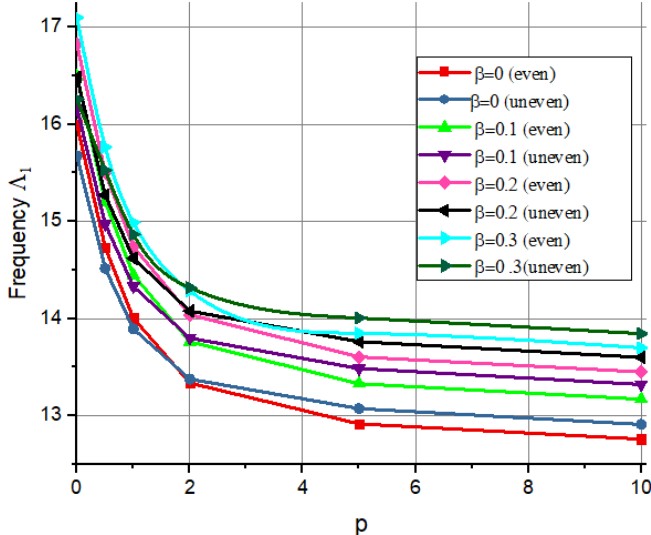

**Figure 5.** Effect of taper ratio $\beta$ and volume fraction index $p$ on natural frequency of clamped FGM plate on elastic foundation ($\alpha = 0.2$, Type I, II, $K_w = 50$, $K_p = 100$).

The frequencies increase with the taper ratio, but they decrease with the increase in the gradient index, which is reasonable for this specific problem. The values of frequency for Type II (uneven distribution) are slightly lower than the frequencies for Type I for all values of the gradient index.

Case 5

An investigation of various types of FG materials is conducted. Two materials are considered in the FGM: $ZrO_2$ and $Ti - 6Al - 4V$; their mechanical properties are indicated as follows:

| | | | |
|---|---|---|---|
| $ZrO_2$: | $E = 200$ GPa, | $\nu = 0.3$, | $\rho = 5700$ kg/m$^3$; |
| $Ti - 6Al - 4V$: | $E = 105.698$ GPa, | $\nu = 0.3$, | $\rho = 4427$ kg/m$^3$. |

Further numerical experiments have been carried out for four FG materials:

- FGM-1: $Al/Al_2O_3$;
- FGM-2: $Al/ZrO_2$;
- FGM-3: $Si_3N_4/SU\ S304$;
- FGM-4: $ZrO_2/Ti$-6 Al-4V.

The FGM plate of Figure 3a is investigated. The variable thickness parameter is $\beta = 0.3$; the plate rests on an elastic foundation ($K_w = 50$, $K_p = 100$) and is clamped (CL-CL). The porosity parameter is $\alpha = 0.2$ for two porosity distribution types—even and uneven; the power-law index is varied as follows $p = 0, 0.5, 1, 2, 5, 10$. The non-dimensional fundamental frequency is reported in Table 9 and Figure 6.

**Table 9.** Effect of the power index on natural frequency of clamped plates on elastic foundation $K_w = 50$, $K_p = 100$, with porosity $\alpha = 0.2$ and variable thickness $\beta = 0.3$, fabricated using different FGMs: FGM-1, FGM-2, FGM-3, FGM-4.

| $p$ | Al/Al$_2$O$_3$ | | Al/ZrO$_2$ | | Si$_3$N$_4$/SU S304 | | ZrO$_2$/Ti-6 Al-4V | |
|---|---|---|---|---|---|---|---|---|
| | **Even** | **Uneven** | **Even** | **Uneven** | **Even** | **Uneven** | **Even** | **Uneven** |
| 0 | 17.0981 | 16.2578 | 11.3984 | 11.2181 | 33.1099 | 28.6338 | 13.4604 | 13.0954 |
| 0.5 | 15.7629 | 15.5251 | 11.7884 | 11.5483 | 20.4428 | 19.4449 | 13.4819 | 13.0915 |
| 1 | 14.9889 | 14.8662 | 12.1286 | 11.8418 | 17.7114 | 17.1402 | 13.5442 | 13.1377 |
| 2 | 14.2821 | 14.3177 | 12.6761 | 12.3071 | 15.7946 | 15.4595 | 13.6646 | 13.2342 |
| 5 | 13.8484 | 14.0049 | 13.5197 | 12.9885 | 14.3382 | 14.1478 | 13.8201 | 13.3554 |
| 10 | 13.6981 | 13.8439 | 13.9614 | 13.3197 | 13.7529 | 13.6123 | 13.8579 | 13.3784 |

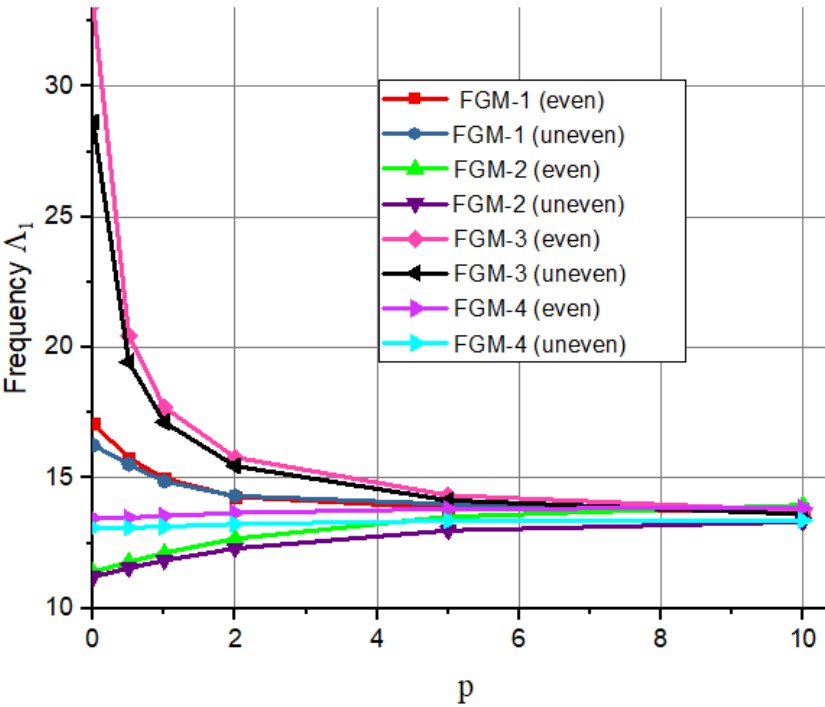

**Figure 6.** Effect of volume fraction index $p$ on natural frequency of clamped FGM plate (different types of materials) on elastic foundation $K_w = 50$, $K_p = 100$ ($\alpha = 0.2$, $\beta = 0.3$).

For the case under consideration, there are several interesting points to discuss. First, as volume fraction index $p$ increases, the frequencies decrease for the FGM-1 and FGM-3 materials, while for the other two, they increase insignificantly. Secondly, for all materials, the frequencies for FGM-2 are less than those for FGM-1. Thirdly, the frequencies for the FGM-3 material are significantly higher than those in other cases. Fourth, the frequencies for the FGM-4 material practically do not change.

Another free vibration analysis with four types of FG materials is fulfilled for clamped (CL-CL) FGM plates with different values of the porosity parameter: $\alpha = 0$, 0.1, 0.2, 0.3, 0.4, 0.5, 0.6. Now, the gradient index is $p = 1$, the taper ratio is $\beta = 0.3$, and the foundation stiffness parameters of Winkler and Pasternak are $K_w = 50$, $K_p = 100$. The non-dimensional fundamental frequencies for even and uneven porosity distribution types are presented in Table 10 and shown in Figure 7.

**Table 10.** Effect of the porosity parameter $\alpha$ of Types I and II on natural frequency of clamped plates, on elastic foundation $K_w = 50$, $K_p = 100$, made of different FGMs with taper ratio $\beta = 0.3$ and gradient index $p = 1$.

| $\alpha$ | Al/Al$_2$O$_3$ | | Al/ZrO$_2$ | | Si$_3$N$_4$/SU S304 | | ZrO$_2$/Ti-6 Al-4V | |
|---|---|---|---|---|---|---|---|---|
| | **Even** | **Uneven** | **Even** | **Uneven** | **Even** | **Uneven** | **Even** | **Uneven** |
| 0 | 14.4329 | 14.4329 | 11.4219 | 11.4219 | 16.4712 | 16.4712 | 12.6371 | 12.6371 |
| 0.1 | 14.6921 | 14.6417 | 11.7449 | 11.6236 | 17.0368 | 16.7923 | 13.0508 | 12.8773 |
| 0.2 | 14.9889 | 14.8662 | 12.1286 | 11.8418 | 17.7114 | 17.1402 | 13.5443 | 13.1377 |
| 0.3 | 15.3278 | 15.1082 | 12.5927 | 12.0789 | 18.5328 | 17.5187 | 14.1448 | 13.4209 |
| 0.4 | 15.7058 | 15.3699 | 13.1659 | 12.3376 | 19.5526 | 17.9325 | 14.8945 | 13.7305 |
| 0.5 | 16.0847 | 15.6541 | 13.8924 | 12.6213 | 20.8891 | 18.3873 | 15.8628 | 14.0707 |
| 0.6 | 16.1709 | 15.9633 | 14.8381 | 12.9341 | 22.6927 | 18.8903 | 17.1731 | 14.4467 |

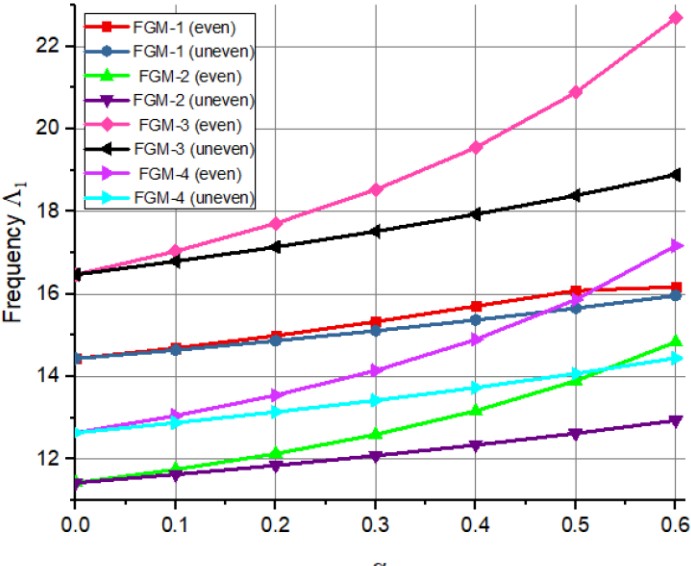

**Figure 7.** Effect of the porosity parameter $\alpha$ of Types I and II on natural frequency of clamped plates, on elastic foundation ($K_w = 50$, $K_p = 100$), made of different FGMs ($\beta = 0.3$, $p = 1$).

It can be noted that with an increase in the porosity parameter, the natural frequencies increase for all FG material types. A common trend is observed for all the materials in a small reduction in the frequency parameter for the uneven case of porosity. Material FGM-3 shows again the biggest values of non-dimensional frequency; frequencies of material FGM-1 are very close for cases of even and uneven porosity types.

## 5. Conclusions

This work considers for the first time the application of the R-functions theory to free vibration problems of porous FGM plates of a complex form, with variable thickness, resting on an elastic foundation. A mathematical formulation of the problem is developed in the framework of the first-order shear deformation theory for FGM plates. It is shown that the application of the R-functions theory together with the Ritz variational method makes it possible to solve a wide range of vibration problems for porous plates. The use of the Ritz variational method allows for taking into account the variability of the plate thickness analytically. A set of admissible functions, constructed using the R-functions theory, exactly satisfies the main boundary conditions, both on the outside region boundary and for holes. The approach was validated in a large number of case studies. New achievements of this work include the following:

(1) The highlight of this work is a demonstration of the effectiveness of the application of the R-functions theory for porous FGM plates of variable thickness resting on an elastic foundation with complex shapes.

(2) The method was used to investigate the free vibration of square plates with clamped holes having a complex form;

(3) A numerical experiment was conducted to study the effects of parameters such as the taper ratio, porosity distribution, foundation stiffness coefficients, volume fraction index, and type of FGM on the natural frequencies.

(4) The application of the R-functions theory is validated by means of a comparative analysis with a traditional finite element approach (COMSOL Multiphysics).

All these issues also allowed the identification of advantages and disadvantages of finite element modeling and boundary condition sensitivity for porous FGM plates with variable thickness.

**Author Contributions:** All authors contributed equally to conceptualization; methodology; validation; software; formal analysis; investigation; resources; writing—original draft preparation; writing—review and editing; visualization; funding acquisition. All authors have read and agreed to the published version of the manuscript.

**Funding:** This research was funded by NATO, project "Composite Metamaterials for Aerospace Structures—CoMetA", grant number G6176, under the framework of the Science for Peace and Security (SPS) Programme.

**Data Availability Statement:** Dataset available on request from the authors.

**Conflicts of Interest:** The authors declare no conflicts of interest.

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
