# Peer review of "Free Vibration Analysis of Porous Functionally Graded Material Plates with Variable Thickness on an Elastic Foundation Using the R-Functions Method"

_mca, doi:10.3390/mca29010010_

Round 1

Reviewer 1 Report

Comments and Suggestions for Authors

This manuscript requires revision to enhance its quality. Below are some proposed amendments:

1. The manuscript's writing needs meticulous editing to improve clarity. The title "Free Vibrations of Porous FGM Plates with Variable Thickness and Resting on Elastic Foundation and the R-Functions Method" lacks clarity and could be more grammatically concise. The issue lies in the repetitive use of "and," which makes the title read like a list of unrelated components. A more coherent version could be: "Analysis of Free Vibrations in Porous FGM Plates with Variable Thickness on an Elastic Foundation Using the R-Functions Method."

2. Merging Figures 1 and 3 would be advisable for better coherence. Moreover, the depiction of the complex hole in Figure 3(a) appears asymmetrical and needs correction.

3. The paper contains equations overloaded with expressions, necessitating division for clarity. For example, Equation (4) encompasses eight formulas, and Equation (11a) three. This issue persists with other equations not specifically mentioned.

4. Consistency in pronoun usage is essential. The majority of the manuscript employs third-person narration, yet some sections unexpectedly switch to first-person, evident in lines 187 and 188: "Let us substitute expressions (25) into (21), resulting in the following representation of the desired solution."

5. A notable discrepancy is observed in Table 5 between the FEM results and those from other methods, particularly for p=5 and alpha=0.2. The reason behind this significant variance warrants explanation.

6. While the uneven distribution of pores is considered, the focus solely on O-type distribution raises the question: why not include an X-type distribution for comprehensive analysis?

7. The solution method described in Section 3 lacks clarity and needs substantial revision. Additionally, certain variables, such as 'varphi' in Equation (25), are not defined, leading to potential confusion.

8. The paper's overall format needs restructuring, as the integration of equations with the text currently lacks fluidity and coherence.

Comments on the Quality of English Language

See the Comments and Suggestions for Authors.

Author Response

Thank you very much for your comments and remarks. We tried to consider almost all of them. Here are our responses.

  1. The manuscript's writing needs meticulous editing to improve clarity. The title "Free Vibrations of Porous FGM Plates with Variable Thickness and Resting on Elastic Foundation and the R-Functions Method" lacks clarity and could be more grammatically concise. The issue lies in the repetitive use of "and," which makes the title read like a list of unrelated components. A more coherent version could be: "Analysis of Free Vibrations in Porous FGM Plates with Variable Thickness on an Elastic Foundation Using the R-Functions Method."

Response. Thank you. We changed the title. Now it is "Free Vibration Analysis of Porous FGM Plates with Variable Thickness on an Elastic Foundation Using the R-Functions Method."

  1. Merging Figures 1 and 3 would be advisable for better coherence. Moreover, the depiction of the complex hole in Figure 3(a) appears asymmetrical and needs correction.

Response. Thank you. From the authors' point of view, Figures 1 and 3 cannot be merged, because Fig. 1 emphasizes that plates of variable thickness are considered. Therefore, a spatial image of the plate is shown. Figure 3 shows the shape of the plate plan and the law of change of its thickness. We corrected Fig. 3, so now it does not appear asymmetrical.

  1. The paper contains equations overloaded with expressions, necessitating division for clarity. For example, Equation (4) encompasses eight formulas, and Equation (11a) three. This issue persists with other equations not specifically mentioned.

Response. Thank you. We wrote equations (4) as (4a) and (4b), and equations (11) as (11a), (11b), (11c), (11d), (11e).

  1. Consistency in pronoun usage is essential. The majority of the manuscript employs third-person narration, yet some sections unexpectedly switch to first-person, evident in lines 187 and 188: "Let us substitute expressions (25) into (21), resulting in the following representation of the desired solution."

Response. Thank you. We corrected this.

  1. A notable discrepancy is observed in Table 5 between the FEM results and those from other methods, particularly for p=5 and alpha=0.2. The reason behind this significant variance warrants explanation.

Response. Thanks for the comment. “In certain instances, such as when p=5, alpha=0.2, and CSCS, a discrepancy arises, which may be attributed to an edge (boundary) layer issue that eludes detection by the FEM, as indicated in Ref. [42].” We added this also in the manuscript.

  1. While the uneven distribution of pores is considered, the focus solely on O-type distribution raises the question: why not include an X-type distribution for comprehensive analysis?

Response. In this work, two laws of porosity distribution defined by the expressions (Type I, even) and (Type II, uneven) are studied. Therefore, the results are presented for these laws (Tables 8-10; Fig. 5-7).

  1. The solution method described in Section 3 lacks clarity and needs substantial revision. Additionally, certain variables, such as 'varphi' in Equation (25), are not defined, leading to potential confusion.

Response. The R-functions method is described quite complete in Refs. [31-34]. In this work, we presented (as an example) the construction of a sequence of admissible functions for completely clamped plate with a complex geometry what is the most complicate case of the boundary conditions.

About the functions  it is said in the text (at the end of page 6). It is some complete system of functions. We have added an example of such a system, used in this work.

Reviewer 2 Report

Comments and Suggestions for Authors

In this paper authors have analysed the free vibrations of functionally graded materials plates. It is assumed that plates have variation in thickness. It is further considered that a two parameter elastic foundation is in contact with plate with acts in tension as well as in compression. Furthermore the effects of material porosity is included into the formulation. The first order shear deformation theory of plates is assumed to estimate the kinematics of the plate  and derive the basic governing equations. The method of R functions is applied to construct the eigenvalue problem. Results of this study are well-compared with the available data in the open literature and after that novel numerical results from present study are provided. This work is well-written and well-organized. However the following comments should be considered in the revised version correctly

1) As I checked the conditions only for the type of fully clamped plates are considered in this researchin Eq. (20). What is the reason? Is it a limitation if the adopted numerical method? Please provide a frank discussuion

2) Authors have used the energy technique to obtain the natural frequencies of the plate. So what is the reason for providing the equations (11) as equations of motion.

3) Please provide the convention for naming the boundary condition. For instance what is CCCS or CFCF?

4) What is the value of shear correction factor in this research? As authors know the value of this factor depends on many parameters such as loading, BCs, geometry and material type. So how do authors have used a value for this factor?

5) A convergence study is needed in this research to examine the convergence of the R-function method

6) Authors may enrich the literature review on the mechanics of FGM plates by considering more works such as [International Journal of Mechanical Sciences 75, 134-144, 2013], [Composite Structures 102, 205-216, 2013], [Acta Mechanica 223 (6), 1199-1218, 2012], [Composite Structures 94 (8), 2474-2484, 2012]

Author Response

Review 2

  1. As I checked the conditions only for the type of fully clamped plates are considered in this research in Eq. (20). What is the reason? Is it a limitation if the adopted numerical method? Please provide a frank discussion.

Response. Authors considered various boundary conditions, both for rectangular plates (Tables 1-5) and for complex geometry (Table 6). Frequency behaviour graphs for complex geometries are indeed presented for clamped plates, including the hole, because it is the most complicated case of the boundary conditions.

  1. Authors have used the energy technique to obtain the natural frequencies of the plate. So what is the reason for providing the equations (11) as equations of motion.

Response. Of course, it would be possible not to present the equations of motion, but for completeness and clarity of the problem statement, we decided to present the equations of motion.

  1. Please provide the convention for naming the boundary condition. For instance, what is CCCS or CFCF?

Response. Thank you. We introduced the corresponding descriptions.

  1. What is the value of shear correction factor in this research? As authors know the value of this factor depends on many parameters such as loading, BCs, geometry and material type. So how do authors have used a value for this factor?

Response. You are absolutely right. But in this work, we used FSDT in the simplest formulation, when the shear correction factor is taken equal to 5/6. Many users of this theory use such value. We hope in the future to use more accurate theories that do not require knowledge of this parameter.

  1. A convergence study is needed in this research to examine the convergence of the R-functions method.

Response. The authors investigated the convergence of the R-functions method by increasing the number of coordinate functions. It was found that stabilization of frequency values in the third digit is achieved for ,  k=1,2,3,4,5… at i=25. That’s the general number of admissible functions is equal to 125. Note that we take into attention the symmetry of the problem. So as complete system  has the following form:

:   ,

  :

  1. Authors may enrich the literature review on the mechanics of FGM plates by considering more works such as [International Journal of Mechanical Sciences 75, 134-144, 2013], [Composite Structures 102, 205-216, 2013], [Acta Mechanica 223 (6), 1199-1218, 2012], [Composite Structures 94 (8), 2474-2484, 2012].

Response. Thank you. We inserted some mentioned works to enrich the literature review.

Reviewer 3 Report

Comments and Suggestions for Authors

My review for the manuscript titled: “Free vibrations of porous FGM plates with variable thickness and resting on elastic foundation and the R-functions method”

I have read the paper carefully. It looks interesting and novel in the field of FGMs. The paper is well-arranged and some sections make everything clear. I checked the mathematical model as well and it is well-derived. However, the manuscript requires some changes and improvements to be suitable for publication. In this regard, the following comments may help,

C1) First, the title can be corrected. “R-function method” made discontinuity in the title. Moreover, there are several elastic foundations, not only the Winkler-Pasternak. I suggest this title: “Free vibrations of porous FGM plates with variable thickness resting on an elastic foundation using R-functions method”

C2) English may be acceptable. However, it needs some corrections. For example,

-Please do not miss “dash” for FSDT. “first-order shear deformation” into the “abstract”.

Introduction:

-“another ones”…shall be…“another one” or “other ones”.

-“It can be noted, that porosity can be occurred in”…shall be…“It can be noted that the porosity can occur in”…occur cannot be passive.

-“researches” is very uncommon cause “research” is a mass noun and should always be written in the singular form.

And more errors which are out of this reviewer’s duty. Please double-check the whole paper.

C3) Functionally graded compositions, theoretically, can be made in the perfect form. However, when it comes to practical and real work, achieving a perfect mixture is somehow impossible. Please discuss this issue using: [Composite Structures, Volume 249, 2020, 112486] to push future readers to think about considering FGMs in imperfect compositions.

C4) Functionally graded compositions can not only impress the mechanical properties, but also it can increase the Multiphysics coupling properties in the smart structures. Thus, it is highly recommended to mix the intelligent materials using FG techniques to obtain better sensing and actuating tools. This case is worth mentioning using: [Continuum Mech. Thermodyn. 34, 1051–1066 (2022)].

C5) Please explain in the introduction how is it practically possible to produce an FGM in such an irregular geometry.

C6) Some units should be corrected. “Gpa” should be “GPa”.

C7) In all the Tables, the decimal of numbers should be the same.

C8) For such a paper with a lot of parameters, I suggest adding a Nomenclature so then that future readers will be able to find any parameter more easily.

C9) Normally, when the materials are composed of FGMs, it is significant to employ a higher-order shear deformation model or a three-dimensional elasticity analysis cause FG plates/beams are usually thick and FSDT does not suit thick domains. This issue can be introduced using: [Composite Structures, Volume 255, 2021, 112925; Mechanics Based Design of Structures and Machines, 50:10, 3596-3625].

C10) The results discussion lacks physical interpretations. For example, Fig. 6. What is the physical reason that in some FGMs while the values of the gradient parameter are increasing from zero, the frequency values are decreasing, while for some others increasing?

C11) According to Fig. 7, it deems that porosity does not have the same effect on the defined FGMs 1-4. Why? Please provide scientific reasoning.

C12) Please add the most important achievement of the work at the end of the abstract.

C13) Please list the new achievements of this research point by point in the "conclusion" section.

Comments on the Quality of English Language

Please see the overall comments.

Author Response

Thank you very much for your comments and remarks. We tried to consider almost all of them. Here are our responses.

  1. First, the title can be corrected. “R-function method” made discontinuity in the title. Moreover, there are several elastic foundations, not only the Winkler-Pasternak. I suggest this title: “Free vibrations of porous FGM plates with variable thickness resting on an elastic foundation using R-functions method”.

Response. We decided to change a little a title. Now it is the following: “Free Vibration Analysis of Porous FGM Plates with Variable Thickness on an Elastic Foundation Using the R-Functions Method”.

  1. English may be acceptable. However, it needs some corrections. For example, -Please do not miss “dash” for FSDT. “first-order shear deformation” into the “abstract”. Introduction: -“another ones”…shall be…“another one” or “other ones”. -“It can be noted, that porosity can be occurred in”…shall be…“It can be noted that the porosity can occur in”…occur cannot be passive. -“researches” is very uncommon cause “research” is a mass noun and should always be written in the singular form. And more errors which are out of this reviewer’s duty. Please double-check the whole paper.

Response. Thank you. We were trying to check twice the paper and correct English grammar.

  1. Functionally graded compositions, theoretically, can be made in the perfect form. However, when it comes to practical and real work, achieving a perfect mixture is somehow impossible. Please discuss this issue using: [Composite Structures, Volume 249, 2020, 112486] to push future readers to think about considering FGMs in imperfect compositions.

Response. Thank you. Taking into account the theme of the work that you mentioned is “A new hyperbolic-polynomial higher-order elasticity theory for mechanics of thick FGM beams with imperfection in the material composition” we can say that discussing issue is interesting, but it is not very close to our problem.

  1. Functionally graded compositions can not only impress the mechanical properties, but also it can increase the Multiphysics coupling properties in the smart structures. Thus, it is highly recommended to mix the intelligent materials using FG techniques to obtain better sensing and actuating tools. This case is worth mentioning using: [Continuum Mech. Thermodyn. 34, 1051–1066 (2022)].

Response. Thank you. Mentioned work “Thermal buckling of functionally graded piezomagnetic micro- and nanobeams presenting the flexomagnetic effect” is very interesting. We will try to use this issue next time.

  1. Please explain in the introduction how is it practically possible to produce an FGM in such an irregular geometry.

Response. The purpose of the work is to propose an effective method for studying the considered plates with different shape, as well as with different hole shape. The production of FGM is an another problem, which can be solved, for example, by means of 3-D printers (from the point of authors' view)

  1. Some units should be corrected. “Gpa” should be “GPa”.

Response. Thank you. We corrected this.

  1. In all the Tables, the decimal of numbers should be the same.

Response. Thank you. We corrected this.

  1. For such a paper with a lot of parameters, I suggest adding a Nomenclature so then that future readers will be able to find any parameter more easily.

Response. Thank you. It is convenient, maybe we will do it next time.

  1. Normally, when the materials are composed of FGMs, it is significant to employ a higher-order shear deformation model or a three-dimensional elasticity analysis cause FG plates/beams are usually thick and FSDT does not suit thick domains. This issue can be introduced using: [Composite Structures, Volume 255, 2021, 112925; Mechanics Based Design of Structures and Machines, 50:10, 3596-3625].

Response. Thank you. We introduced this issue. Please, find Ref. 39.

  1. The results discussion lacks physical interpretations. For example, Fig. 6. What is the physical reason that in some FGMs while the values of the gradient parameter are increasing from zero, the frequency values are decreasing, while for some others increasing?

Response. The behaviour of FGM frequencies with increasing gradient index depends on the properties of the constituent materials. If we compare the values of the modulus of elasticity and density of ZrO2 and Al2O3 we can see that they differ significantly. Therefore, the behaviour of frequencies for these materials can be qualitatively different.

  1. According to Fig. 7, it seems that porosity does not have the same effect on the defined FGMs 1-4. Why? Please provide scientific reasoning.

Response. Figure 7 shows the results for a fixed value of p=1. Therefore, it does not follow from Fig. 7 that the porosity coefficient does not affect the behaviour of frequencies for different types of FGM.

  1. Please add the most important achievement of the work at the end of the abstract.

Response. According to the general scheme of this article the most important achievement was indicated in Conclusion. Please, you can find it there.

  1. Please list the new achievements of this research point by point in the "conclusion" section.

Response. Thank you. We did it.

Round 2

Reviewer 1 Report

Comments and Suggestions for Authors

After careful review and revisions, the paper now meets the required standards and is ready for acceptance.

Reviewer 2 Report

Comments and Suggestions for Authors

The revised version is satisfactory and may be accepted in its current form